# Micromirror-Embedded Coverslip Assembly for Bidirectional Microscopic Imaging

**DOI:** 10.3390/mi11060582

**Published:** 2020-06-10

**Authors:** Dongwoo Lee, Jihye Kim, Eunjoo Song, Ji-Young Jeong, Eun-chae Jeon, Pilhan Kim, Wonhee Lee

**Affiliations:** 1Graduate School of Nanoscience and Technology, Korea Advanced Institute of Science and Technology (KAIST), Daejeon 34141, Korea; ldw6871@kaist.ac.kr (D.L.); xcv4482@kaist.ac.kr (J.K.); sej_cosmos@kaist.ac.kr (E.S.); pilhan.kim@kaist.ac.kr (P.K.); 2Department of Nano Manufacturing Technology, Korea Institute of Machinery & Materials (KIMM), Daejeon 34103, Korea; nano-jeong@kimm.re.kr; 3School of Materials Science and Engineering, University of Ulsan, Ulsan 44776, Korea; jeonec@ulsan.ac.kr; 4Graduate School of Medical Science and Engineering, Korea Advanced Institute of Science and Technology (KAIST), Daejeon 34141, Korea; 5Department of Physics, Graduate School of Nanoscience and Technology, Korea Advanced Institute of Science and Technology (KAIST), Daejeon 34141, Korea; 6Department of Bio and Brain Engineering, Korea Advanced Institute of Science and Technology (KAIST), Daejeon 34141, Korea

**Keywords:** micromirror, microfabrication, 3D imaging, Optical MEMS

## Abstract

3D imaging of a biological sample provides information about cellular and subcellular structures that are important in cell biology and related diseases. However, most 3D imaging systems, such as confocal and tomographic microscopy systems, are complex and expensive. Here, we developed a quasi-3D imaging tool that is compatible with most conventional microscopes by integrating micromirrors and microchannel structures on coverslips to provide bidirectional imaging. Microfabricated micromirrors had a precisely 45° reflection angle and optically clean reflective surfaces with high reflectance over 95%. The micromirrors were embedded on coverslips that could be assembled as a microchannel structure. We demonstrated that this simple disposable device allows a conventional microscope to perform bidirectional imaging with simple control of a focal plane. Images of microbeads and cells under bright-field and fluorescent microscopy show that the device can provide a quick analysis of 3D information, such as 3D positions and subcellular structures.

## 1. Introduction

3D imaging of biological samples provides invaluable information about cellular and subcellular structures, which is indispensable for cell biology and pathophysiology of related diseases [1,2,3,4,5,6,7]. Therefore, many 3D imaging tools for life science have been developed. Among them, the most common tool is undoubtedly confocal microscopy. Confocal microscopy reconstructs 3D information with an optical sectioning technique that captures multiple 2D images: fluorescent molecules and a spatial pinhole are used to image small specimen volume and to reject out-of-focus light [8,9,10]. Various techniques, such as 4pi, structured illumination, and plane illumination microscopy, have been developed to achieve high contrast and resolution that are close to those of electron microscopy [11]. Nonetheless, it is relatively time-consuming and high in cost. In addition, confocal microscopy systems have low temporal resolution and cannot offer their maximal spatial resolution for dynamic targets.

Various tomographic methods have also been widely used for 3D imaging [4,12,13]. For example, Optical diffraction tomography (ODT) is a powerful imaging tool that can provide 3D images of a refractive index (RI) distribution within samples, which are reconstructed from multiple 2D images of the samples [14,15,16,17,18]. Although the tomographic methods have various advantages, such as label-free, quantitative, and fast 3D imaging capabilities, they require a high-precision optomechanical or optoelectronic device to acquire multiple 2D images of samples from multiple angles. Various microfluidics and microfabrication techniques have been developed to overcome the limitations in implementing tomography to applications in life science. For example, cells were imaged while they were rolling in a microchannel flow [19]. We have recently developed a novel ODT technique using micromirror-embedded coverslips [20]. Micromirror-embedded coverslips allowed simultaneous imaging of the sample in orthogonal directions, which led to the enhancement of the 3D reconstruction performance of ODT without sample rotation or illumination scanning.

Micro-optical components have been integrated with microfluidic devices to achieve unconventional imaging/sensing systems for various bio-applications [21,22]. The micromirror-embedded coverslip devices can be used in conventional microscopy, enabling quasi-3D bioimaging in a single field of view (Figure 1). The coverslips have micropatterns that can be assembled to form a microchannel that guides the loading of samples by capillary wetting. Two micromirrors are integrated side by side to divert the optical path and to allow the side view imaging. The assembled device is compatible with high magnification objective lenses that have small working distances. High compatibility and ease of use will allow this system to help to acquire 3D information of bio-samples without additional, complex optical setup. For example, 3D cell culture needs analysis of complex 3D interactions between cells [23,24,25]. Under the micromirror-embedded coverslip, cellular structure, distribution of specific molecules, and cell motility in 3D can be easily observed. Here, we first describe the details of the fabrication processes of the micromirror-embedded coverslips. We then show that the micromirror-embedded coverslips can allow robust and convenient acquisition of 3D information of biological samples under conventional bright-field and fluorescent microscopy systems.

## 2. Materials and Methods

### 2.1. Preparation of a PDMS Channel for Micromirror Capillary Molding

A brass block cut by a planing process used as a master mold for the micromirrors. Polydimethylsiloxane (PDMS) (Dow Corning, Midland, MI, USA) base and cross-linker were mixed in 1:10 ratio and poured on the surface of the brass. For an easy detachment of PDMS from the brass, detergent (MICRO-90, International Products Corporation, Burlington, NJ, USA) was sprayed on the surface of the brass mold and dried before pouring uncured PDMS. The uncured PDMS on the brass was degassed in a desiccator for 30 min and cured for 2 h at 65 °C. The cured PDMS was detached from the brass, which was used for the replica mold of a PDMS channel. The PDMS replica mold was placed in a Petri dish and coated with the detergent. Then, uncured PDMS was poured into the petri dish. The PDMS was cured in the same way as that described above. The PDMS channel was finally cut and detached from the mold.

### 2.2. Fabrication of Coverslip with Parallel Column Structures

Molds for the parallel column structures were fabricated by conventional photolithography using a SU-8 photoresist (SU-8 3050, MicroChem, Westborough, MA, USA). A Si wafer was coated with the SU-8 photoresist with 1300 rpm for 40 s. After the spin-coating, the wafer was baked for 30 min at 95 °C. Then, UV light was exposed to pattern the parallel column structures with 9.1 mW for 26 s using a mask aligner (MA-6). After the exposure, the wafer was baked for 1 min and 10 min at 65 °C and 95 °C, respectively. Then, the wafer was developed using SU-8 developer (MicroChem) for 15 min, and baked for 5 min at 150 °C. The final height of the SU-8 pattern was 100 μm. After the SU-8 patterning on the Si wafer, uncured PDMS was mixed and poured on the Si wafer and degassed in the desiccator for 30 min. Then, it was cured for 2 h at 65 °C. The cured PDMS channel was cut and detached from the SU-8 mold. The channel inlets and outlets were punched, and the PDMS channels were placed on a coverslip without plasma bonding. Then, UV-curable polymer (NOA71, Norland, Fort Atkinson, WI, USA) was filled in the channels by a capillary force and cured for 30 min under UV lamp (365 nm, UVITEC CAMBRIDGE, Cambridge, UK). Then, after curing the UV-curable polymer, the final coverslip was prepared by the detachment of the PDMS mold and cutting the excessive pieces at the inlets and outlets of the PDMS channels.

### 2.3. Preparation of Beads and Cells

Some 10 μm fluorescent beads (Thermo fluoro-max, excitation 468 nm, and emission 508 nm) were used for measurement. The solvent in the bead solution was dried, and the remaining beads were dispersed in UV-curable polymer (NOA71). Then, the capillary channel was filled with the bead solution by capillary action.

H460-H2B-GFP cells that express a histone H2B-GFP in nuclei were used. In addition, cells were fluorescently labeled by 30 min of incubation with a red lipophilic fluorescent dye, Vybrant DiD cell-labeling solution (Thermo Fisher Scientific, V22887, Waltham, MA, USA). After washing with Dulbecco’s Phosphate-Buffered Saline (DPBS), labeled cells were suspended in Matrigel solution (10^4^ cells∙mL^−1^, Matrigel 10–50 v/v% in DPBS) and injected into a microfluidic channel. Red blood cells were taken from a healthy donator and diluted 3000-fold with DPBS.

## 3. Results and Discussion

### 3.1. Fabrication of Trapezoidal Micromirror and Micromirror-Embedded Coverslip Assembly

Micromirrors were fabricated by various microfabrication techniques, including soft-lithography and sputter deposition of a metallic reflective coating (Figure 2a). A polymeric microprism structure was made by capillary molding from a microchannel mold; then, silver as a reflective layer was sputter-coated on surfaces of the microprism. We investigated several fabrication methods for the microchannel mold because the quality of the micro-optical structure is directly determined by the microchannels. To construct the microchannel with the desired geometry and optical quality, two important requirements were considered: precise control of the angles of the sidewalls and optically smooth mirror surfaces. It is not a simple task to meet such requirements with a microchannel structure constructed by conventional wet and dry etching techniques. The etch profile of anisotropic wet etching of Si can be tuned by controlling the concentrations of additives such as alcohol [26,27]. Sidewall angles of 45° can be achieved using a tetramethylammonium hydroxide (TMAH) and isopropyl alcohol (IPA) mixture as an etchant [28]. This method creates a microchannel mold with a cross-sectional shape of the equilateral right triangle. However, it results in rough large side walls, and the resulting sidewalls are not suitable for mirror surfaces. To overcome the roughness problem, the hypotenuse side of the triangle was constructed by assembling the microchannel on an optically smooth substrate, and the bottom surface was used as a mirror surface, which made fabrication steps complicated. In addition, the sidewall angle was very sensitive to etching conditions, such as temperature and stirring speed, resulting in relatively low reproducibility. Here, we constructed the master mold by a planing process (i in Figure 2a). The planing process is a high-precision machining technique that uses diamond cutting-tools to cut out a metal workpiece (Figure 2b) [29,30]. The planing process can provide a structure with precise angles and optically smooth surfaces [31,32,33]. We used a diamond tip of trapezoidal shape and built the cross-section of micromirror as a trapezoid with a side angle of 45° (Figure 2b inset), which allowed easy manipulation and positioning of the micromirrors in the embedding process. The master mold was constructed on a bulk brass plate.

A PDMS mold was replicated from the brass mold (ii in Figure 2a). The replicated PDMS mold was used for the molding of PDMS channels with a trapezoidal cross-section (iii in Figure 2a). Figure 2c shows the cross-section of the PDMS channel, having a trapezoidal shape with precise angles. The dimensions of the trapezoidal cross-section are base:height = 250:200 μm. The PDMS channels were assembled with a flat PDMS sheet to construct microchannels (iv in Figure 2a). UV-curable polymer (OG142, Epoxy Technology) was filled in the channels and cured under a UV lamp (365 nm, UVITEC CAMBRIDGE) for 30 min (v in Figure 2a). The PDMS microchannel could be easily detached from the cured micro-structures because oxygen inhibition of the UV-curing process leaves a thin layer of uncured UV-curable polymer [34,35]. The top PDMS channel was highly flexible and could be slowly peeled off from one side. The microstructure could remain on the bottom PDMS sheet. Then, 150-nm-thick silver was deposited as a reflective coating layer on the inclined sides using a DC sputter (vi in Figure 2a). The fabricated micromirror structure can be cut into pieces with a length of 5–10 mm (Figure 2d). Salt grains are placed in between the two mirrors to show the reflectance of silver. Silver has a high reflectivity in the visible spectrum and is widely used as a reflective coating for mirrors. However, silver can be easily oxidized, depending on the surrounding environment. The micromirror directly contacts liquids with high salt concentrations, such as cell culture media, buffer solutions, and blood. In the previous research, we used aluminum as a reflective coating because aluminum is chemically stable due to the native oxide layer. However, aluminum has inferior reflectance compared to silver. We used silver as the reflective coating and deposited a passivation layer of a 1-μm-thick parylene layer using a commercial parylene coater (Labcoater 2, Specialty Coating Systems, Figure 2e). Parylene has very low permeability to a wide range of gases, including water vapor, so it is often used as a protective coating for electronics and medical tools. In addition, parylene is transparent in a wide spectral range and can be conformally coated as a thin film of micrometer to nanometer thickness, making it a good candidate for the passivation material of the micromirror.

A micromirror-embedded coverslip assembly consists of two coverslips with different sizes (22 mm × 22 mm and 24 mm × 50 mm, ~120–170 μm thickness), where the trapezoidal micromirror is embedded on each coverslip (Figure 3). Two types of parallel column structures were designed and fabricated using capillary molding (Figure 3a). The PDMS mold was replicated from the SU-8 mold fabricated by conventional photolithography using SU-8 photoresist (i in Figure 3a, and Figure 3b, MicroChem). The parallel column structure has a gap distance of 1100 μm and a height of 100 μm, which is designed for subsequent alignment of the micromirrors and the two coverslips. After the PDMS molding process of the SU-8 pattern, the parallel column structures were patterned on each coverslip by capillary molding of UV-curable polymer (ii and iii in Figure 3a, and Figure 3c, NOA71, Norland). The micromirror was aligned next to the column structure and fixed using the UV-curable polymer as an adhesive (iv in Figure 3a). Finally, the two micromirror-embedded coverslips were aligned and assembled (v in Figure 3a, and Figure 3d). The coverslip can be permanently bonded with UV-curable polymer or simply stacked together depending on applications.

### 3.2. Bidirectional Imaging of Fluorescent Beads with the Micromirror-Embedded Coverslip

The quality of images reflected from the micromirrors was evaluated with 10 μm fluorescent beads (Figure 4). The beads were dispersed in a UV-curable polymer solution. The solution was loaded into the microchannel by capillary filling, and the solution was cured by UV exposure so that the position of the beads would not change. Imaging of beads and cells was acquired using an inverted optical microscope (Nikon Eclipse Ti-U, Nikon, Tokyo, Japan) and a fluorescent illuminator (Nikon Intenslight C-HGFIE) system. The 20× objective lens with a numerical aperture of 0.45 was used. We examined the side-view images for distortion of the shape or changes in fluorescent intensity compared to the images in top view. The fluorescent images of the beads were captured from the top-view (original image) and side-views (reflected image from the micromirrors) after focusing on the individual beads from each side (Appendix A). To distinguish the top-view and side-view images of an object, we use the location of mirror 2 as the reference. An object in the side-view was found within the location of the mirror 2. In addition, the focal plane for an object in the side view is further away than the focal plane of an object in a top-view that lies between the top and bottom glass slides. Four representative images show almost identical circular shapes from both top-views and side-views (Figure 4a). Figure 4b shows the bright-field images of bead 1 for comparison.

Fluorescent intensity profiles were plotted along the center of the bead in the horizontal and the vertical directions to compare the top-view and the side-view images of bead 1 (Figure 4c,d). An analysis of fluorescent intensity was conducted with Image J software. The measured intensity ratios of side-view/top-view are over 95% at the maximum values. Note the emission wavelength of the fluorescent bead is 508 nm, and the reflectance of the thin-film silver coating is typically over 90% at a wavelength of 500 nm. The intensity ratios from all the fluorescent beads were larger than 95%, showing superior quality of the micromirror (Appendix A, Table 1). The reflectance of the reflective coating is expected to be uniform over the entire micromirror area. It was found that there was a slight variation of the reflected fluorescent intensity depending on the position of the beads in the channels. The numerical aperture (NA) for the side-view can be limited by the height of the mirror (250 μm); therefore, the intensity of the reflected image would be reduced if the bead was located far away from the micromirror. The bead 3 was located relatively further away from the mirror surface, which gave a slightly smaller intensity ratio.

In addition to the reflectance, we have evaluated whether the reflected images have any distortions by the mirror. Roundness was measured to show how close the images were to perfect circles. The roundness is calculated by *4A/π*(*Major axis*)*^2^*. Here, *A* is the area of a circular image, and the major axis is the longest diameter of the circular image. A value close to 1 means that the object is close to a perfect circle. The values of the roundness for the side-view images are smaller than those from the top-view images, which suggests that there exists minor elongation of the reflected images. The aspect ratio is defined as vertical length/horizontal length. The aspect ratio also shows that bead images in the top view are almost perfect circles, while the reflected images are slightly elongated in the z-direction. Figure 4d also shows that the bead has a slightly larger size in the vertical intensity profile. The evaluation of roundness and the aspect ratio indicates that the side-view images have tiny deviation (less than a few %) from a perfect circle. The bead 3 again has relatively large (~ 5%) distortion, which could be the result of the limited NA.

### 3.3. Bidirectional Imaging of Live Cells with the Micromirror-Embedded Coverslip

Live-cell imaging under the micromirror-embedded coverslip devices was demonstrated using bright-field and fluorescent microscopy. H460, human large-cell lung carcinoma, cells that express a histone H2B-green fluorescence protein (GFP) in nuclei were used for both setups. In addition, the cell membrane was fluorescently labeled with a red lipophilic fluorescent dye. The cells were prepared in Matrigel solution (10^4^ cells∙mL^−1^, Matrigel 10–50 v/v% in DPBS) and loaded in the microchannel by capillary wetting. After the cell loading, the Matrigel solution was hardened within a few minutes. Figure 5 shows the images taken under a conventional bright-field microscope (Nikon Eclipse Ti−U). The 20× objective lens with a numerical aperture of 0.45 was used. Images of the cell in a top-view and a side-view typically are not focused simultaneously; clear images of cells were obtained by simply changing the focal plane (Figure 5b). Figure 5c shows a region of high cell concentration. The cells could be imaged clearly at high magnification (40× objective lens) in both top-view and side-view. The results clearly demonstrate that the images taken in the two orthogonal views can provide 3D information on the morphology of cells and positions of the cells (Figure 5d).

In addition to the H460 cells, the red blood cells (RBCs) were imaged, with which bi-directional imaging of cell morphology can have practical importance (Figure 5e). The 50× objective lens with a numerical aperture of 0.55 was used. The biconcave disc shape morphology of the RBCs could be clearly imaged from top-view and side-view. RBC 1 and RBC 2 show the change in morphology as time goes on in a PBS buffer; the RBC 2 has shrunk by osmotic pressure. Here, RBCs are suspended in a PBS, and they settle down to the bottom of the channel. An additional structure of 100 μm height was fabricated in the capillary channel to keep the cells in the middle of the channel cross-section (in the z-direction) (Appendix A).

Bidirectional images under fluorescent microscopy can provide 3D subcellular structural information without confocal scanning. In addition, the intensity of the fluorescence can provide quantitative as well as qualitative information. Fluorescent microscopic imaging of a 3D subcellular structure is demonstrated with a micro-mirror embedded coverslip device using a fluorescent illuminator (Nikon Intenslight C-HGFIE) system (Figure 6). The membranes (red) and nuclei (green) of H460 cells were imaged from top-view and side-view. The cell membrane was fluorescently labeled with Vybrant DiD, and the loading of the cells into the microchannel was done using the same steps as previously described. The results clearly show that the detailed 3D shape of the subcellular structure can be quickly identified under bidirectional imaging with a simple conventional microscopy system. If an appropriate cell culture system is integrated into the microfluidic channel, various cellular activities, such as cell division or cell migration in a 3D cell culture platform, could be observed.

The use of additional mirrors orthogonal to existing mirrors could provide another side-view direction, which would lead to 3-direction imaging, and it could provide more information. However, the additional mirrors would make it difficult to construct a microchannel for the injection of samples. For 3-direction imaging, the microfluidic system can take a multiple-well structure instead of a channel structure. In this case, samples can be dispensed in the wells using a pipette.

## 4. Conclusions

We developed a micromirror-embedded coverslip device for bidirectional imaging that is compatible with conventional optical microscope systems without additional equipment. The micromirrors with a 45° angle reflection surface were integrated on coverslips. The micromirrors were fabricated through several microfabrication processes, including the planing process, soft lithography, and metal deposition. The planing process with a diamond cutting tool provided a precise angle and optically flat and clean reflection surface. The assembled coverslips have a capillary microchannel, allowing easy sample loading. The fidelity of the reflected image was evaluated by comparing the top-view and side-view images of fluorescent microbeads. The ratio of fluorescent intensities from direct and reflected images was over 95%, which was owing to the high reflectance of silver reflective coating that was passivated by a thin-film parylene protective layer. We have demonstrated the acquisition of 3D information bio-samples using live cells. The morphology and subcellular structure from top-views and side-views could be imaged with high magnification, and their 3D positions could be easily measured. Although the device does not provide the benefit of the higher resolution of confocal microscopy, the device can provide simple quasi-3D imaging, and it is highly compatible with a conventional bright-field and fluorescent microscope system. We also anticipate this device can be utilized for advanced imaging techniques, such as multiphoton and light-sheet microscopy to simplify the optical setup.

## Figures and Tables

**Figure 1 micromachines-11-00582-f001:**
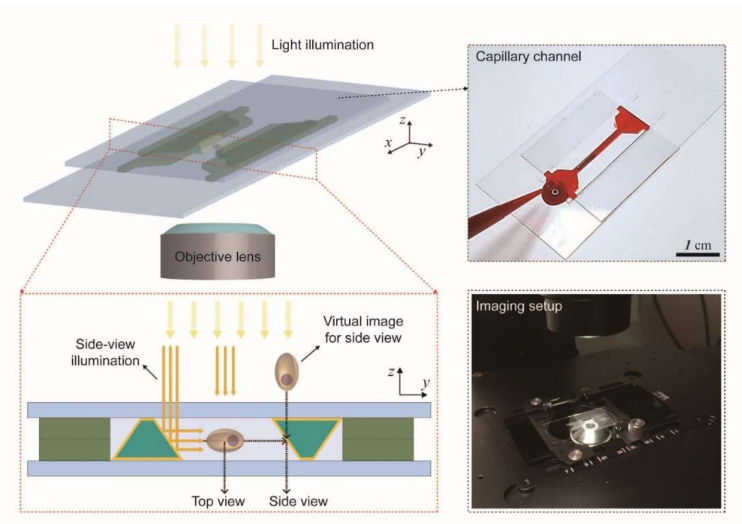
Schematics for bidirectional imaging with a micromirror-embedded coverslip assembly. A microfabricated capillary channel and micromirrors are integrated on coverslips to provide top-view and side-view images of a sample at the same time with a conventional microscope system.

**Figure 2 micromachines-11-00582-f002:**
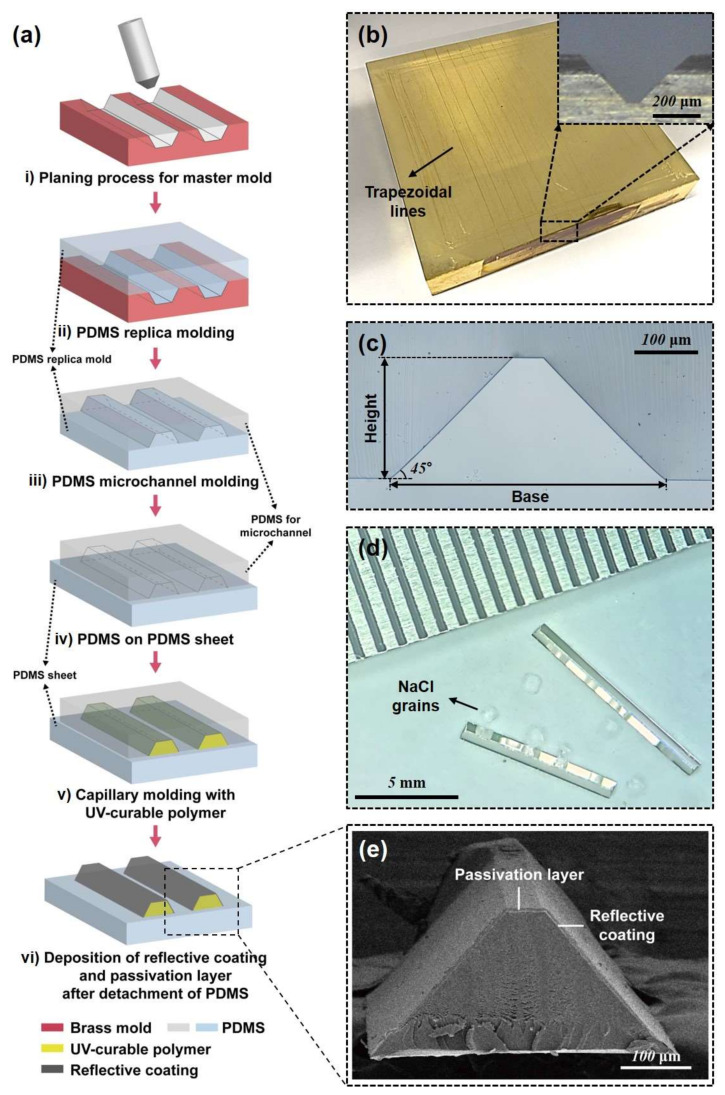
A fabrication process for a micromirror. (**a**) Fabrication steps for trapezoidal micromirror. (**b**) Brass mater mold with a trapezoidal cross-section. (**c**) Cross-section view of PDMS channel. (**d**) Optical microscopic image of micromirrors with grains of salt. (**e**) SEM images of a micromirror.

**Figure 3 micromachines-11-00582-f003:**
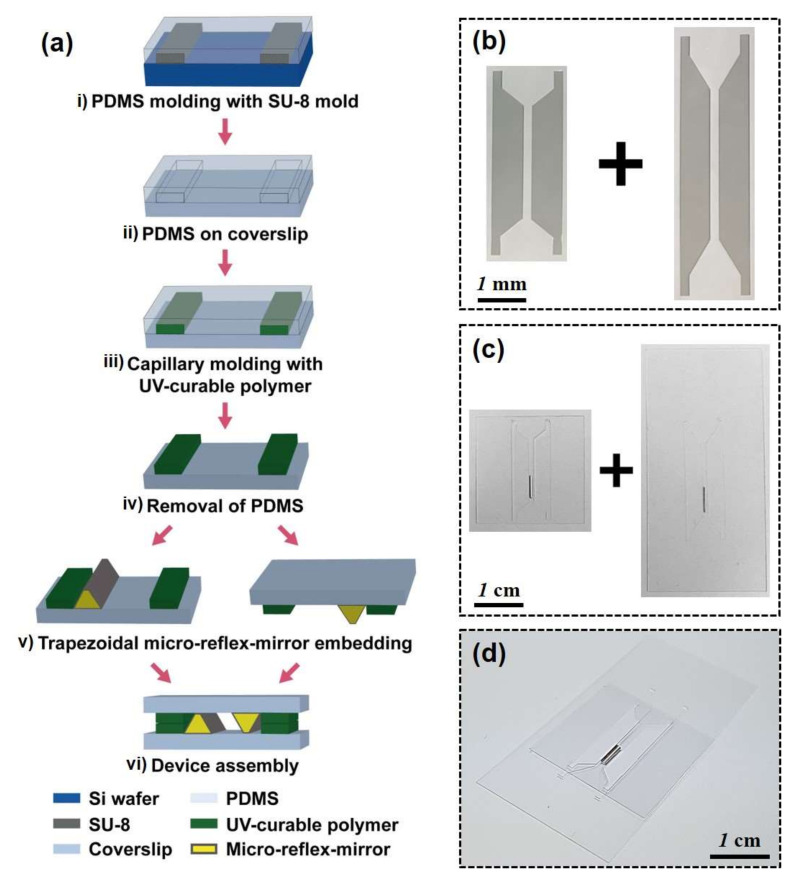
A fabrication process for micromirror-embedded coverslip assembly. (**a**) Fabrication steps for coverslip assembly. (**b**) SU-8 mold for parallel column structures (**c**) Coverslips with the embedded micromirrors (**d**) Final assembled coverslip device.

**Figure 4 micromachines-11-00582-f004:**
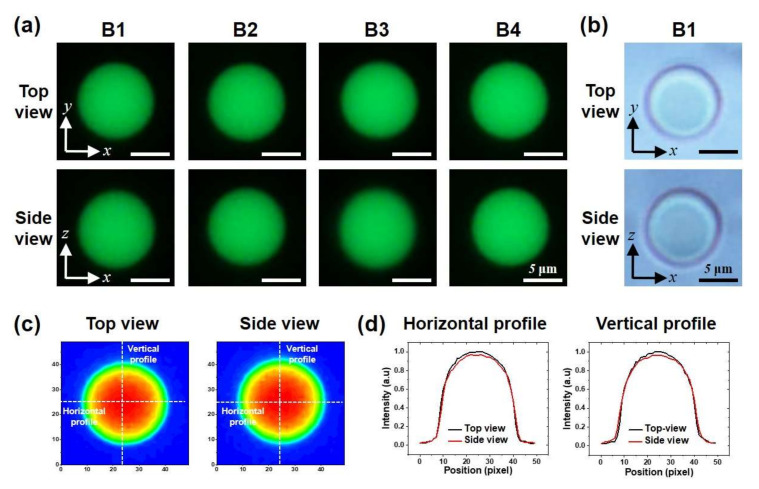
Evaluation of bidirectional imaging with fluorescent beads. (**a**) Fluorescent microscopy images of four beads from top-view and side-view. (**b**) Bright-field images of bead 1. (**c**) Fluorescent intensity contours of bead 1 and (**d**) its intensity profile through the center lines in horizontal and vertical directions.

**Figure 5 micromachines-11-00582-f005:**
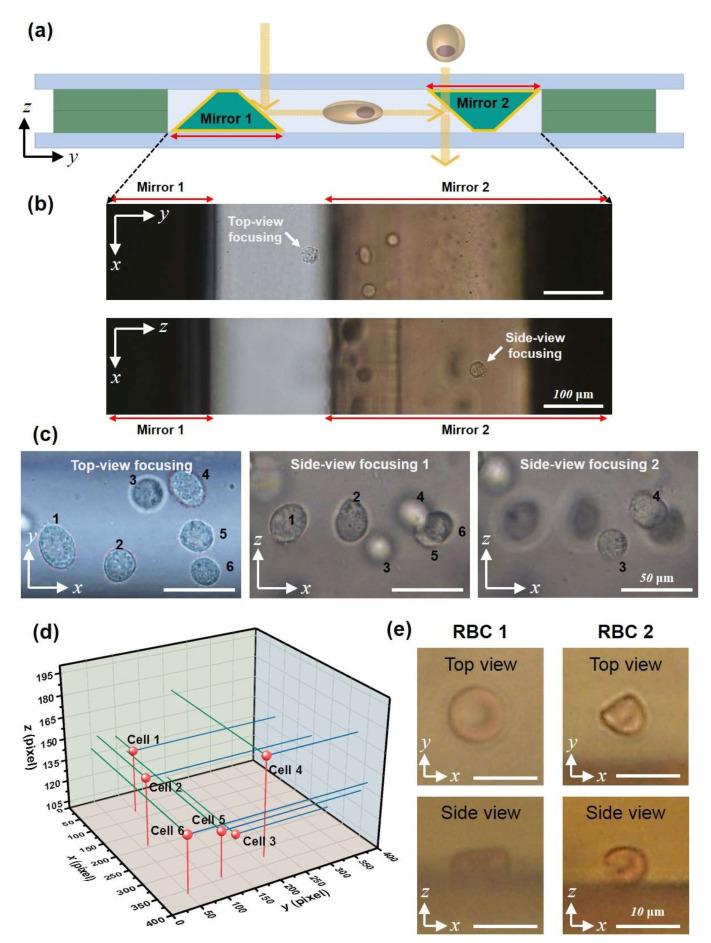
Quasi-3D imaging of live cells with a micromirror-embedded coverslip. (**a**) Schematic of bidirectional imaging. (**b**) Corresponding bright-field microscopy images of an H460 cell (with arrows, obtained with 20× objective lens). The cell is located at different focal planes in the top-view and the side-view. (**c**) Image of multiple H460-GFP cells from top-view and two side-views with focusing at different depths. (**d**) 3D positions of the cells from c. (**e**) Bright-field images from the top-view and side-view of red blood cells (obtained with 50× objective lens).

**Figure 6 micromachines-11-00582-f006:**
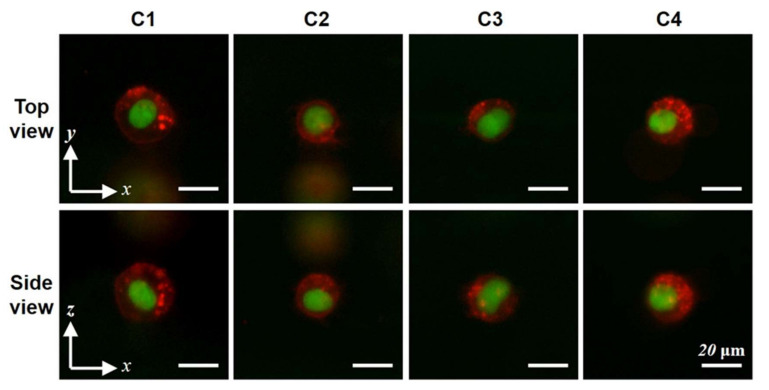
Fluorescent images of subcellular structure from top-view and side-view. Top-view and side-view images (obtained with 40× objective lens) of four H460 cells with the fluorescent-labeled cell membrane (red) and nucleus (green).

**Table 1 micromachines-11-00582-t001:** Comparison of the direct and reflected images of fluorescent beads.

	Intensity Ratio		Roundness		Aspect Ratio		Distance from the Mirror (μm)
	Horizontal	Vertical	Top	Side	Top	Side	
B1	0.97	0.963	0.998	0.971	1.002	1.03	43.4
B2	0.978	0.978	0.998	0.988	1.002	1.012	17.1
B3	0.957	0.957	0.994	0.951	1.006	1.052	97.5
B4	0.972	0.979	0.994	0.981	1.006	1.019	20.7

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
