# Peer review of "Micromirror-Embedded Coverslip Assembly for Bidirectional Microscopic Imaging"

_micromachines, 2020, doi:10.3390/mi11060582_

Round 1

Reviewer 1 Report

The manuscript proposes an innovative design consisting of micromirrors on a glass coverslip to achieve bidirectional imaging using a conventional microscope set-up. The authors successfully show both, top-view and side view images of fluorescent particles, live cells using their design. Overall, the manuscript is well-written and provides convincing results that demonstrate the capability of the proposed design to provide bidirectional imaging. A simple and easy way to enhance imaging capabilities of a simple microscope might be of great interest to a wide research community using microscopes for various imaging tasks and therefore, I would recommend publication of the manuscript with minor revisions. Some comments to improve the manuscript are as follows.

Comments:

  1. In Figure2, I would recommend authors to name the materials and label the schematic for easier understanding of the fabrication process.
  2. Line 130: Which base of the trapezoid is 250 microns? It would be nice to label it in the schematic to avoid confusion.
  3. Line111-124. The whole text in this section is very confusing. Did the authors try different techniques for fabricating the micromirror and found metal planing as the most suitable technique for preparing the mold for the micromirror? If yes, please make the statements concise and clear. Just mention the techniques briefly and the limitation of each techniques in achieving the desired structure.
  4. Line 138-139. What do the authors mean by channel assembled with flat PDMS sheet. Do they bond the two PDMS together. The authors should also show this step in the schematic . It is difficult to correlate this step with the fabrication schematic in Figure 2a.
  5. Line140-141: Since both the top and bottom material are PDMS, why does the cured epoxy easily detach only from the PDMS with microchannel but stick to the PDMS sheet.
  6. In Figure 2D, it appears that the silver film is deposited on only one of the inclined surface of the micromirror and is uneven. Is this true? Please explain briefly in the text the image shown in the figure 2d. What do the bright spots correspond to? Also place a scalebar in the figure?
  7. Figure 3a, the PDMS mold obtained should be the inverse of the SU-8 mold. The schematic looks identical to that of the SU-8 mold. Instead of pillars, the PDMS mold should have channels. If this is the case, please correct the schematic. The schematic is confusing. Labelling of materials, dimensions etc. would make the schematic more informative.
  8. How was the micromirror aligned next to the column on the coverslip. Was this done manually? Since the spacing between column is only 1000 micron, it would require high precision to place the micromirror accurately. Explain the process.
  9. For Figure 4, it is enough to show the fluorescence image of only one bead (B1), the rest can be put in the supplementary figures.
  10. The manuscript should be corrected for grammatical and spelling errors. For example, in Figure 4, “pixcel” should be labelled as pixel.
  11. Is there a specific requirement on the type of objective lens that can be used for side imaging using this set-up. For example, are there any restriction on the working distance, numerical aperture of the objective lens that need to be considered for proper operation of the side imaging in the current set-up. A comment on this would be helpful.
  12. It would be nice to see images (both top-view and side view) using different objective lens with different magnification and working distances.
  13. How were the top-view and side-view images distinguished in the CCD image. Explain the process in detail and also describe how the position of cells were obtained in Figure 5d.
  14. In Figure 5b, label both the images to correlate with the schematic shown in Figure 5a. Also include the coordinate axis.
  15. Line223-224. The authors mentioned both bright-field and fluorescent images, but no fluorescent images of the cells are shown in Figure 5.
  16. Line 237-238. What capability ? looks something is missing?
  17. Line 93, “ PDMS channels were put on a coverslip.” Does it mean bonded or simply put on top a coverslip. In the latter case, would the UV-epoxy flow out of the channels while capillary molding.
  18. The side view image in supplementary information, Figure V1 is missing.

Author Response

We appreciate the reviewer’s comments. We made changes in the manuscript to address the reviewer’s questions and suggestions. Please see the attachment.

Reviewer 2 Report

This is an interesting study showing how to embed mirrors for imaging of microscopic objects from the side as well as in direct transmission imaging. I would suggest the authors to clarify and consider the following points:

It is stated that the technique is "similar to confocal" (abstract line 25). Yet the system is not at all confocal. This is a false claim and should be corrected. The system has simultaneous bi-directional imaging but no confocality. Confocality allow sectioning imaging and improved transverse resolution. This is not the case here.

As the mirrors (micro, but almost macro) take op space adjacent to the structure being imaged there is a risk that the effective NA will be reduced. Please comment.

When imaging adjacent cells simultaneously some could cause shadowing in the side-view mode, but not the forward mode. Please comment.

Would it be advantageous to use 3 or 4 rather than 2 mirrors to allow imaging from all sides? Also, could the system be easily modified for fluorescent reflection imaging, rather than transmission? These would be a good point to discuss.

The English is for the most clear throughout and I found just one disturbing typo: remained beads were dispersed -> remaining beads were dispersed (line 100, section 2.3)

Author Response

(The authors gave the same response as above.)

Reviewer 3 Report

The paper presented a novel type of microscopic coverslip which allows the simultaneous imaging of top view and side view of cell-sized objects without requiring any additional modification of the microscope instrument itself. Overall, the proposed design demonstrated creative thinking and detailed implementation. The paper is well-organized into logical sections, which provide comprehensive overview of the fabrication, experiment design and result analysis. There are a few points, however, that could be further improved:

  1. In the introduction section, the authors have compared the advantages of the proposed method over confocal microscopy. It may also be necessary to discuss/compare the resolution difference between the two types 3D imaging technique. Typically, confocal microscopy can achieve sub-micron level of resolution. Judging by the nature of the proposed design, it will be very challenging to obtain the same level of resolution. Nonetheless, as the main proposed application of the proposed design is for cell imaging, this disadvantage does not undermine the value of the proposed design. A more comprehensive comparison may help the readers to better understand the limit of the proposed design.
  2. Figure 1 in Section 1 is somewhat hard to interpret. The authors may consider re-do the plots to improve readability.
  3. Typically, the term “micro-mirror” refers to a special type of MEMS device with controllable/movable mirror surface. Admittedly, there is no rule forbidding the use of “micro-mirror” for the more general definition of “small sized reflective surface”, but for certain group of readers the paper title could be misleading. I would suggest using “micro-reflex-mirror” instead of “micro-mirror” to avoid confusion.

Author Response

(The authors gave the same response as above.)

Reviewer 4 Report

In this paper, the author present a method to get  a quasi-3D information on several microns size objects (10µm fluorescent beads, and cells) using a clever micromirror approach, combined with a simple microscope.

The micromirror system was already published by the same group, with tomography measurements and is cited in the current paper, as well as the use of a diamond planing tool for the master mold fabrication. The novelty is thus low, and concerns only the characteristics of the mirror working in fluorescence and white light illumination.

One major problem is that the fabrication description is no new, but appears both in section 2 and section 3, as part of the results.

All the fabrication description should be included in section 2.

Also in section 2, lots of step are quickly described without referring to the Figures of the article, which makes the fabrication process hard to understand. Giving a bit more details in section 3 is then further confusing.

A bit more information should be given in the figures, and maybe another naming, for example microfluidic channel instead of "parallel structures".

Line 177, the used microscope objective is not described, magnification and numerical aperture should appear here.

Some minor english changes :

Line 178 replace with "We have examined on the side-view images if there is a distortion of the shape or changes in fluorescent intensity compared to the images in top view."

line 241-242, the following sentence is not clear: "An imaging stage of 100 μm height was fabricated in the capillary channel to keep the cells in the middle of the channel cross-section (in the z-direction)". The term Imaging stage is not appropriate and should be changed.

line249- 250 : "the cells loading in the microchannel was followed the same steps as previous" should be replaced by "The loading of the cells into the microchannel was made using the same steps as previously described."

In conlusion, the paper should be re-ordered and clarified before publication.

A discussion on the resolution limit and more details on NA limitation could add some interest to the paper.

Author Response

(The authors gave the same response as above.)

Round 2

Reviewer 4 Report

The authors responded well to all concerns previously expressed.